# Adherence to Diet Quality Indices and Breast Cancer Risk in the Italian ORDET Cohort

**DOI:** 10.3390/nu16081187

**Published:** 2024-04-17

**Authors:** Martina Quartiroli, Chiara Roncallo, Valeria Pala, Vittorio Simeon, Fulvio Ricceri, Elisabetta Venturelli, Lara Pattaroni, Sabina Sieri, Claudia Agnoli

**Affiliations:** 1Epidemiology and Prevention Unit, Fondazione IRCCS Istituto Nazionale dei Tumori, 20133 Milan, Italy; martina.quartiroli@istitutotumori.mi.it (M.Q.); chiara.roncallo@istitutotumori.mi.it (C.R.); valeria.pala@istitutotumori.mi.it (V.P.); lara.pattaroni@istitutotumori.mi.it (L.P.); claudia.agnoli@istitutotumori.mi.it (C.A.); 2Dipartimento di Salute Mentale e Fisica e Medicina Preventiva, Vanvitelli University, 80138 Naples, Italy; vittorio.simeon@unicampania.it; 3Department of Clinical and Biological Sciences, Centre for Biostatistics, Epidemiology, and Public Health, University of Turin, 10126 Turin, Italy; fulvio.ricceri@unito.it; 4Nutritional and Metabolomic Research Unit, Fondazione IRCCS Istituto Nazionale dei Tumori, 20133 Milan, Italy; elisabetta.venturelli@istitutotumori.mi.it

**Keywords:** dietary patterns, breast cancer, chronic low-grade inflammation, prospective cohort

## Abstract

Breast cancer (BC) is the most common cancer in women, with 2.3 million diagnoses in 2020. There is growing evidence that lifestyle factors, including dietary factors, particularly the complex interactions and synergies between different foods and nutrients (and not a single nutrient or food), may be associated with a higher risk of BC. The aim of this work was to evaluate how the Italian Mediterranean Index (IMI), the Greek Mediterranean Index, the DASH score, and the EAT-Lancet score can help lower the risk of BC, and analyze if chronic low-grade inflammation may be one of the possible mechanisms through which dietary patterns influence breast cancer risk. We evaluated the effect of adherence to these four dietary quality indices in the 9144 women of the ORDET cohort who completed a dietary questionnaire. The effect of adherence to dietary patterns on chronic inflammation biomarkers was evaluated on a subsample of 552 participants. Hazard ratios (HRs) with 95% confidence intervals (CIs) for BC risk in relation to the index score categories used were estimated using multivariable Cox models adjusted for potential confounders. Regression coefficients (β), with 95% CI for C-reactive protein (CRP), TNF-α, IL-6, leptin, and adiponectin levels in relation to adherence to dietary patterns were evaluated with the linear regression model adjusted for potential confounders. IMI was inversely associated with BC in all women (HR: 0.76, 95% CI: 0.60–0.97, P trend = 0.04), particularly among postmenopausal women (HR: 0.64, 95% CI: 0.42–0.98, P trend = 0.11). None of the other dietary patterns was associated with BC risk. Higher IMI and Greek Mediterranean Index scores were inversely associated with circulating CRP (β: −0.10, 95% CI: −0.18, −0.02, and β: −0.13, 95% CI: −0.21, −0.04). The higher score of the EAT-Lancet Index was instead associated with a higher concentration of circulating levels of CRP (β: 0.10, 95% CI: 0.02, 0.18). In conclusion, these results suggest that adherence to a typical Italian Mediterranean diet protects against BC development, especially among postmenopausal women, possibly through modulation of chronic low-grade inflammation.

## 1. Introduction

Breast cancer is the most prevalent cancer in women. Breast cancer incidence has increased rapidly, especially in some areas, e.g., Australia/New Zealand, North America, and North Europe [1], with 2.3 million diagnoses in 2020 [2]. Breast cancer is the cause of about 685,000 deaths per year and it is the fifth leading cause of cancer-related death in women [1]. In 2023, in Italy, breast cancer was the most frequently diagnosed cancer in women (55,900 cases), followed by colorectal (50,500), lung (44,000), and bladder (29,700) [3]. The high incidence of breast cancer in developed countries reflects a greater prevalence of reproductive and hormonal risk factors (early age at menarche, advanced age at menopause, advanced age at first birth, fewer children, less breastfeeding, hormone therapy in menopause and oral contraceptives) and risk factors related to lifestyle (alcohol intake, excess body weight, physical inactivity), as well as greater early diagnosis through mammographic screening [2].

There are several studies that demonstrated how factors such as adiposity, physical activity, and diet are important risk factors for the development of cancers, including breast cancer [4,5]. Recommendations in the last World Cancer Research Fund (WCRF) report highlighted the following as essential factors to reduce risk of developing cancer: breastfeeding, staying slim, being physically active; following a plant-based diet, rich in nonstarchy vegetables and foods containing carotenoids and rich in calcium, whole grains and legumes; limiting the consumption of red and processed meat; limiting the consumption of processed foods, sugary drinks, and alcohol [6].

However, evidence linking specific foods or nutrients to breast cancer risk are limited, mainly reporting a higher risk due to increased alcoholic drinks intake [6]; part of this inconsistency may be explained by the fact that people do not eat isolated foods but a complex combination of foods and nutrients, and the joint action may be more important than that of any single food or nutrient. In recent years, in nutritional epidemiology, an approach to evaluate the relationship between diet and the onset of diseases, including tumors, based on the analysis of dietary patterns has emerged. Rather than focusing on a single nutrient, dietary patterns might be a more adequate approach to explain the link between diet and cancer. Dietary patterns consider how foods and nutrients are consumed in combination; thus, they capture cumulative and interactive effects among dietary components, providing a comprehensive assessment of diet and reflecting real-world dietary preferences [7,8].

In several studies, dietary patterns have been studied in their relation to breast cancer. In particular, the Mediterranean diet, a diet based on a high consumption of plant-based foods, especially whole grain products, vegetables, fruit, nuts, and legumes, moderate to high consumption of fish, low consumption of eggs, a limited consumption of sugary drinks, red and processed meats, milk, butter, dairy products, and sweets, and a low content of saturated fatty acids (SFA) (7–8% of energy), with total fat between 25% and 35% of the total energy [9], have been associated with a protective effect on several cancer types, including breast cancer [10]. Moreover, the Dietary Approach to Stop Hypertension (DASH) diet, initially recommended for management of hypertension [11], and characterized by a high intakes of fruits, vegetables, whole grains, nuts, and legumes, moderate intakes of low-fat dairy products, and low intakes of sodium, sugar-sweetened beverages, and red and processed meats [12], has been inversely associated with several noncommunicable diseases [13,14], such as cardiovascular diseases, and different types of cancer, including breast cancer [15]. In addition to the concept that diet as a whole may benefit our health, a dietary model that would also be sustainable from an environmental point of view has recently received increasing attention. The EAT-Lancet diet was proposed as a healthy universal reference diet that could reduce the incidence of diet-related obesity and noncommunicable diseases, such as cardiovascular diseases, cancer, and diabetes. EAT-Lancet is based on 2500 kcal/day and characterized by a high consumption of whole grains, fruits, vegetables, nuts, legumes, unsaturated oils, low to moderate amounts of fish and poultry, and no or low red and processed meat, added sugars, refined grains, and starchy vegetables [16]. However, no study until now has evaluated the effect of the EAT-Lancet diet on the development of breast cancer.

The aim of this study was to evaluate if the Mediterranean diet, the DASH diet, and the EAT-Lancet diet may help prevent breast cancer, and to analyze whether chronic low-grade inflammation may be one of the possible mechanisms through which dietary patterns influence the risk of this cancer among the women recruited in the Italian ORDET prospective cohort.

## 2. Materials and Methods

### 2.1. Study Population

The ORDET cohort consisted of 10,786 healthy women, residents of Varese Province in northern Italy, recruited between June 1987 and June 1992. Age at recruitment was 35–69 years, excluding women taking hormone therapy in the three months before recruitment, with chronic or acute liver disease, with a history of cancer, pregnant or breastfeeding women, or women who had undergone bilateral ovariectomy. The study protocol was approved by the Ethics Committee of Fondazione IRCCS Istituto Nazionale dei Tumori, Milan, in Italy. The study complies with the Helsinki Declaration, and participants gave written informed consent to use clinical data for research. At recruitment, information on lifestyle, menstrual history, and reproductive history was collected; height, weight, and waist and hip circumferences were measured; and blood and urine samples were collected. Women also completed a self-administered semiquantitative food frequency questionnaire (FFQ) [17]. The FFQ only became available 30 months after starting recruitment. Women recruited at the beginning (n = 1547) who did not complete the FFQ were excluded from the present analysis. We also excluded women found to have a cancer diagnosis before recruitment (n = 37) or lost to follow-up immediately after recruitment (n = 58), leaving 9144 potentially eligible women for analysis on cancer risk. We subsequently excluded women for whom variables used as covariates in the statistical model were missing, and those for whom the ratio of total energy intake (determined from the FFQ) to basal metabolic rate (determined by the Harris–Benedict equation) [18] was in the first or last half-percentiles of the distribution, to reduce the effect of implausible extreme values on the analysis.

In order to evaluate the association of diet scores with inflammatory markers, we used a subsample of ORDET women who gave blood twice and participated in a nested case–control study on chronic low-grade inflammation and breast cancer risk. For these women, therefore, two plasma samples were available: one collected during the ORDET recruitment and a subsequent one on average five years later. In this subset, 276 breast cancer cases and 276 controls—matched for age (±5 years, date of recruitment (±180 days), and menopausal status (postmenopausal, premenopausal, and perimenopausal at baseline)) were included.

### 2.2. Food Frequency Questionnaire

Dietary habits over the preceding year were assessed using a validated semiquantitative FFQ, as extensively described elsewhere [17]. Volunteers completed the FFQ, on their own, at recruitment, with immediate review by a nurse/volunteer so as to draw attention to any missing items. The FFQ consisted of 107 food items and included photos with two or three sample dishes of definite sizes, or references to standard portion sizes. Participants could specify the frequency of consumption of items by day, week, or month. Questions on seasoning and food preparation were also included. The composition in nutrients of individual food items was obtained from Italian food composition tables [19] and average intakes of macro- and micronutrients for each volunteer were estimated.

### 2.3. Diet Quality Indices

The Italian Mediterranean Index (IMI) score was developed to adapt the Greek Mediterranean Diet score [20] to Italian eating behavior [21]. It is calculated from intake of 11 items: high intakes of six typical Mediterranean foods [pasta, typical Mediterranean vegetables (raw tomatoes, leafy vegetables, onion and garlic, salad, fruiting vegetables), fruit, legumes, olive oil, and fish], low intakes of four “non-Mediterranean” foods (soft drinks, butter, red meat, and potatoes), and alcohol. If consumption of typical Mediterranean foods is in the third tertile of the distribution, the person receives 1 point; all other intakes receive 0 points. If consumption of non-Mediterranean foods is in the first tertile of the distribution, the person receives 1 point. Alcohol receives 1 point for intake up to 12 g/day; abstainers and persons who consume more than 12 g/day receive 0. Since intake of soft drinks was not investigated in the ORDET FFQ, possible scores range from 0 to 10.

The Greek Mediterranean Index is based on the Mediterranean diet scale [20]. Scoring is based on the intake of nine items: vegetables, legumes, fruit and nuts, dairy products, cereals, meat and meat products, fish, alcohol, and the ratio of monounsaturated to saturated fat. For most items, consumption above study median receives 1 point; all other intakes receive 0 points. For dairy products, meat, and meat products, consumption below the median receives 1 point. Medians are gender-specific. For ethanol, men who consumed 10–50 g/day and women who consumed 5–25 g/day received 1 point; otherwise, the score was 0. Since nuts intake was not investigated in the ORDET FFQ, we excluded nuts from the calculation of the “fruit and nuts” component. The range of possible scores is 0 to 9.

We scored DASH as suggested by Fung et al. [22]. The score reflects an individual’s adherence to the DASH diet, which reduces blood pressure and LDL-cholesterol [23,24]. The score is based on eight components: high intake of fruits, vegetables, nuts and legumes, low-fat dairy products, and whole grains, and low intake of sodium, sweetened beverages, and red and processed meats. Subjects were classified into quintiles according to intake of each component. High scores for fruits, vegetables, nuts and legumes, low-fat dairy products, and whole grains indicated high consumption, high scores for sodium, red and processed meats, and sweetened beverages reflected lower consumption. The ORDET FFQ did not contain questions on intake of soft drinks, nuts, and whole cereals; thus, we (a) excluded nuts from the calculation of the “nuts and legumes components”, (b) did not calculate the “soft drinks” component, and (c) replaced the “whole grains” component with fiber from cereals. Component scores were summed to obtain the DASH diet score (range 7 to 35).

The EAT-Lancet score describes adherence to a healthy reference diet proposed in 2019 by the EAT-Lancet Commission, with the aim of being environmentally sustainable and preventing diet-related chronic diseases and mortality [16]. The construction of the EAT-Lancet diet score has been previously described elsewhere [25]. Briefly, it was based on the intake of 14 components, with a possible score of 0 or 1: whole grains (1 point if cereals intake ≤ 464 g/d and cereal fiber > 5 g/d); potatoes (1 point for intake ≤ 100 g/d); vegetables (1 point for intake ≥ 200 g/d); fruits (1 point for intake ≥ 100 g/d); milk and dairy foods (1 point for intake ≤500 g/d); red and processed meat (1 point for intake ≤ 28 g/d); poultry (1 point for intake ≤ 58 g/d); eggs (1 point for intake ≤ 25 g/d); fish (1 point for intake ≤ 100 g/d); legumes (1 point for intake ≤ 100 g/d); soy foods; nuts; unsaturated to saturated fat ratio (1 point for intake ≥ 0.8); sweeteners (i.e., sugars from cake, 1 point for intake ≤ 31 g/d). We could not include soy foods and nuts; thus the score ranges from 0 to 12.

### 2.4. Inflammatory Biomarkers

Measurements of inflammatory markers in plasma samples collected five years after recruitment and adiponectin and C-reactive protein (CRP) of plasma collected at recruitment were performed in 2017 by Luminex multiplex technology, using antibody kits from Bio-Rad, Hercules, CA, USA (TNF-α, IL-6, leptin, and adiponectin) or from the Merck company, Rahway, NJ, USA (CRP) [26]. The inflammatory markers TNF-α, IL-6, and leptin in plasma collected at recruitment were instead assayed as part of a project recently funded by the Italian Ministry of Health using the Human Magnetic Luminex^®^ Assay kits from the Biotechne R&D company (Minneapolis, MN, USA) on a Luminex “Magpix” model instrument. These biomarkers were already available from a previous nested case–control study on chronic low-grade inflammation and breast cancer risk [26] and another recently funded study that aims to evaluate the role of diet in chronic low-grade inflammation and risk of cancer and other chronic degenerative diseases.

### 2.5. Follow-Up

In order to ascertain incident cancer cases up to 31 December 2012, the ORDET database was linked to the local Varese Cancer Registry, which is considered a high-quality registry with less than 3% of cancers identified only through death certificates [27]. Cancer cases were coded according to the 10th edition of the International Classification of Diseases. For analysis on breast cancer risk, participants were followed from study entry until first cancer diagnosis (except nonmelanoma skin cancer), death, emigration, or end of follow-up, whichever occurred first. After a median follow-up of 22.6 years, 587 breast cancer cases were diagnosed.

### 2.6. Statistical Analysis

All baseline characteristics are reported by tertiles of the IMI score. Continuous variables are presented as means with SDs. Categorical variables are presented as counts and percentages. Multivariate Cox proportional hazard models were used to assess the association of diet scores with cancer risk. We ran a minimally adjusted model, with age (continuous) and nonalcoholic energy intake (continuous) as covariates, and a fully adjusted model, further adjusted for age at menarche (continuous), menopausal status (premenopausal/perimenopausal/postmenopausal/not classified), parity (nulliparous/1–2 sons/>2 sons), age at first birth (≤20 years/>20–≤25 years/>25 years), smoking status (never/former/current), education (≤8 years, >8 years), and BMI (continuous). In all models, age was the primary time variable. Hazard ratios (HRs) with 95% confidence intervals (Cis) were estimated for tertiles of diet scores (based on whole population), with lowest tertile as reference. Linearity of trends across tertiles was tested by treating each tertile as a continuous variable in the Cox model. We also ran models for the whole cohort and separately for postmenopausal and premenopausal women.

In order to analyze whether dietary patterns were associated with inflammatory markers CRP, TNF-α, IL-6, leptin, and adiponectin, we used linear regression modeling. We used a model adjusted for age, menopausal status (premenopause/perimenopause/postmenopause/unclassified), and for the time distance between recruitment and conduct of the nested case–control study [26]. Logarithmic transformation was applied to the inflammatory marker values as they were not normally distributed. Since in the two studies the inflammatory markers dosages were obtained with different instruments and kits, we standardized the variables before calculating the difference; lastly, we calculated the differences and standardized them. The analyses were performed with Stata version 18 (College Station, TX, USA).

## 3. Results

### 3.1. Baseline Characteristics of ORDET Women

Table 1 shows baseline characteristics of participants according to tertile of IMI score. Nonalcoholic energy intake increased with increasing adherence to IMI. Moreover, women in the highest tertile of adherence were more likely to be premenopausal, older at first birth, and more educated. 

### 3.2. Associations between Diet Scores and Breast Cancer Risk

Table 2 shows significantly reduced breast cancer risk for increasing adherence to the IMI in all women [HR: 0.76 (95% CI: 0.60–0.97) for the third vs. first tertile, P-trend = 0.04 in the fully adjusted model]. The protection found is linear. As adherence to the IMI increases, the protection becomes stronger.

When we analyzed postmenopausal and premenopausal women separately, the inverse association was stronger among postmenopausal women [HR: 0.64 (95% CI: 0.42–0.98) for the third vs. first tertile]; however, P-trend (0.11) was not significant. The relationship is not linear. In fact, we have a slight nonsignificant increase in risk in the second tertile [HR: 1.07 (95% CI: 0.80–1.43)] followed by a strong and significant protection in the third tertile [HR: 0.64 (95% CI: 0.42–0.98)]. The trend in this case is not linear, but looks more like an inverted parable.

No association was found for the IMI among premenopausal women. No association was found between the Greek Mediterranean Index, DASH diet, EAT-Lancet score, and breast cancer risk.

### 3.3. Associations between Diet Score with the Inflammatory Markers

Table 3 shows the β coefficients of the analysis of the association of diet scores with the inflammatory markers CRP, TNF-α, IL-6, leptin, and adiponectin. It was observed that a greater adherence to the IMI and the Greek Mediterranean Index by the participants is associated with a lower concentration of circulating levels of the inflammatory marker CRP [β: −0.10 (95% CI: −0.18, −0.02)] and [β: −0.13 (95% CI: −0.21, −0.04)].

Greater adherence to the EAT-Lancet Index was instead associated with a greater concentration of circulating levels of the inflammatory marker CRP [β: 0.10 (95% CI: 0.02, 0.18)]. We did not find any significant association with circulating levels of inflammatory markers and adherence to the DASH diet.

## 4. Discussion

In this study, we evaluated the association between four different dietary quality indices and breast cancer risk. The Greek Mediterranean Index and the IMI were chosen to describe adherence to the universally recognized healthy dietary tradition of the Mediterranean diet: the Greek Mediterranean Index because it one of the most applied scores in epidemiological studies (especially among European populations) to describe how adherence to the Mediterranean diet may affect chronic–degenerative disease risk [20]; and the IMI because it was developed to adapt the Greek Mediterranean Diet score to Italian eating behavior [21].

We then decided to calculate the DASH diet, as an example of another healthy dietary pattern specifically designed to reduce chronic disease risk. Indeed, DASH was initially designed to evaluate the risk of developing cardiovascular diseases, but was later found to also have an important role in the oncology field. Finally, a sustainable food model has been receiving increasing attention, including from an environmental point of view, which is why we decided to use the EAT-Lancet index, which was designed to act as a reference point for integrating sustainability into the national dietary recommendations of culturally different countries.

We also evaluated the association between the diet scores and the plasma levels of five inflammatory markers already measured for previous funded studies [26]. Previous studies [28,29] found that increased concentration of these inflammatory markers was associated with increased risk of different types of cancer, and in particular with breast cancer. Furthermore, there is evidence that dietary components can modulate these markers [30,31].

The results of the present study show a significant reduction in the risk of developing breast cancer as adherence to the IMI increases in all women. The inverse association is stronger among postmenopausal women. The results of a 2017 systematic review [10] show how greater adherence to a Mediterranean-type dietary model is inversely associated with the risk of developing breast cancer. Furthermore, an intervention study of 2015 [32] showed that the risk of developing breast cancer decreased in postmenopausal women who followed a Mediterranean-type diet. A probable reason why we did not find significant results also for premenopausal women could be due to the fact that, for women who are not yet in menopause, the onset of the disease is more influenced by genetic factors, for example, a greater prevalence of BRCA1 and BRCA2 mutations [33], or by exposures to other factors during the early stages of adult life. Evidence exists that the main risk factors for premenopausal breast cancer include reproductive variables such as nulliparity and oral contraceptive use—i.e., factors linked to hormonal exposure [33].

Thus, the effect of diet on breast cancer risk among premenopausal women may be greatly surpassed by the effect of these genetic and reproductive factors.

We observed that greater adherence to the IMI and the Greek Mediterranean Index was associated with a lower concentration of the inflammatory marker CRP; instead, a higher EAT-Lancet Index score was associated with a higher concentration of the inflammatory marker. In a number of systematic reviews and meta-analyses [10,34,35], Mediterranean type dietary patterns were associated with lower concentrations of inflammatory biomarkers such as CRP and TNF-α.

Our results of a null association between adherence to the DASH diet and breast cancer risk are not in line with the results of two systematic reviews [36,37], showing how greater adherence to the DASH dietary model was inversely associated with the risk of developing breast cancer [38]. This may be due to the fact that the DASH diet has traditionally been studied for the American population, whose eating habits and dietary survey and assessment tools are different from those used in this study. The DASH index score is a little different from that used in other studies because the ORDET FFQ did not contain questions on intake of soft drinks, nuts, and whole cereals; thus, we excluded nuts from the calculation of the “nuts and legumes components”, did not calculate the “sweetened beverages” component, and replaced the “whole grains” component with fiber from cereals. Furthermore, compared to other studies, we used different covariates for the analyses: for example, we did not use physical activity, because there were no specific questions in the questionnaire, nor hormonal therapies, because the women who were following these therapies were excluded from the study.

A recently published manuscript evaluated the association between eight a priori-defined dietary patterns and major chronic diseases, defined as a composite outcome of incident major cardiovascular diseases, type II diabetes, and cancer [39]. Among other scores, the authors evaluated the effect of DASH and the AMED, an adaptation of the principles of the traditional MD to non-Mediterranean countries [40]. Results for total cancer showed a significant decreased risk for higher adherence to both dietary indices; however, the decreased risk was greater for AMED compared to DASH, especially for obesity-related cancers. The authors also evaluated associations between individual food groups and major chronic diseases, showing that these associations were weaker than those found for dietary patterns, thus highlighting that dietary patterns could reflect the overall effects of diet beyond the sum of individual foods.

In our study, no significant association was found between adherence to the EAT-Lancet diet and the risk of developing breast cancer. Surprisingly, we also found that a higher score of the EAT-Lancet Index was associated with a greater concentration of circulating levels of the inflammatory marker CRP. These results could derive from the fact that when the ORDET study was designed, the importance of the environmental sustainability of the diet had not emerged yet. Consequently, the ORDET FFQ may not be a suitable instrument to capture this aspect of the diet. Indeed, to calculate adherence to EAT-Lancet score, we had to change the original version because not all components were available from the ORDET FFQ. To the best of our knowledge, only one manuscript evaluated the possible relationship between EAT-Lancet score and chronic low-grade inflammation [41]. In this study, the authors evaluated the association between change in adherence to the Mediterranean Diet Score and to the EAT-Lancet score and change in plasma CRP and chemerin levels in a subsample of the EPIC-Potsdam cohort who gave blood and dietary information more than once between 1994–1998 and 2013. Results showed that stable high or increasing adherence to both indices compared to stable low adherence was associated with slight and no significant reduction in CRP and chemerin. Concerning the association between adherence to EAT-Lancet score and breast cancer development, no study has been published until now.

A probable reason why we did not find significant results on breast cancer risk also for the other indices could be due to the fact that the IMI, as it is constructed, better captures the eating habits of the Italian population and may be able to better characterize its healthier aspects than the other indices.

The IMI consider pasta, characterized by a medium/low GI, as a positive component and potatoes, characterized by a high GI, as a negative component among sources of carbohydrates. Instead, for building the other indices, we included total cereals or cereal fiber: this could prevent capturing the beneficial effect of lowering diet GI on breast cancer. For the vegetable component, we decided to consider typical Mediterranean vegetables (raw tomato, leafy vegetables, onion and garlic, salad, fruiting vegetables) and for fat sources, we did not consider the ratio between monounsaturated and saturated fats, as for the Greek Mediterranean Index. We decided to use butter as a negative component and olive oil as a positive component: this may have enabled us to capture the healthy effect of all its components rather than that of fatty acids alone.

The slightly better performance of the Italian compared to the Greek Index might be due to the use of tertiles to classify consumption of items in the former, with the “health” point awarded only if the quantity consumed was in the highest tertile. The Greek Index used the median as cutoff between healthy and nonhealthy consumption. Thus, our Italian Index, compared to the Greek, gives more weight to high adherence to a Mediterranean diet.

For the construction of the IMI, we decided to also consider alcohol consumption, as our objective was to evaluate the eating habits of the Italian population, who have the habit of drinking 1–2 glasses of wine with a meal. However, unlike the Greek Mediterranean Index, in which alcohol component was scored positively when ethanol consumption was from 5 to 25 g/day, for the IMI, we chose to reduce alcohol consumption of up to 12 g/day for scoring this component positively. Since alcohol has been associated with increased breast cancer risk, this may contribute to the better performance of the IMI compared to the Greek [6].

Thus, our findings support a protective role of the Mediterranean diet in the development of breast cancer. There are several biological mechanisms through which the Mediterranean diet can be considered a protective factor against breast cancer. The Mediterranean dietary model is characterized by high proportions of fruit and vegetables, an excellent source of antioxidants and other micronutrients that help prevent oxidative damage to cells, being able to neutralize free radicals by donating an electron or hydrogen atom to a wide range of reactive oxygen, nitrogen, and chlorine species; can prevent the activation of carcinogens, suppress spontaneous mutations by preventing DNA damage, inhibit metastasis and proteases produced by tumor cells, exhibit antiproliferative properties, leading to the downregulation of PI3K (phosphatidylinositol 3-kinase) proliferation pathways, MAPK (mitogen-activated protein kinase), and NF-κB (nuclear factor kappa-light-chain-enhancer of activated B cells), and induce apoptosis [42,43,44].

Flavonoids exert prominent antioxidant and anti-inflammatory activities through various mechanisms. In addition to their role in food intake regulation and nutrition absorption, a growing body of evidence supports that flavonoids increase adiponectin levels and AMPK activation and counteract NF-κB and inducible nitric oxide synthase (iNOS) signaling pathways, resulting in reduced oxidative damage and inflammation associated with obesity [45,46,47,48].

Several studies [49] have demonstrated that polyphenols of the Mediterranean diet exert a direct effect on autophagy. The effects of resveratrol on autophagy might be explained by its enhancing effect on the activity of deacetylase sirtuin 1 [50], which in turn regulates the activity of several autophagy-related proteins. Likewise, polyphenols present in virgin olive oil, such as oleocathal and oleuropein, have been reported to enhance autophagy [51,52].

Lycopene, a phytochemical carotene contained in tomatoes, a characteristic food of the Mediterranean diet and, in particular, of the IMI, induces the upregulation of the activity of antioxidant enzymes (SOD, GPX, and catalase) and shows anti-inflammatory and sensitizing properties to insulin [53,54].

The bioactive components of extra virgin olive oil, as shown in the PREDIMED intervention study [32], appear to have endothelium-protective and antioxidant properties, improving the concentration of inflammation markers such as C-reactive protein, IL-6, and those related to endothelial function such as flow-mediated dilation and E-selectin [51,52].

Fibers in fruit and vegetables have a protective role against breast cancer development as they are able to lower the level of circulating estrogen concentrations through the inhibition of intestinal reabsorption of estrogens excreted in the bile and by increasing their fecal excretion [55].

The Mediterranean diet causes a significant increase in plasma levels of sex hormone binding globulin and insulin-like growth factor binding protein 1 and 2, which reduce the biological activity of estradiol, insulin-like growth factor 1, and testosterone [56]. Moreover, estrogenic molecules with low potency, such as biochanin A, formononetin, daidzein, coumestans, and genistein found in beans, can compete with the endogenous estrogens for binding to estrogen receptors, hence blocking their mitogenic effects [57,58].

Dietary fiber, together with the consumption of low glycemic index foods, characteristic of the Mediterranean diet, can decrease insulin resistance [59], leading to a reduction in insulin levels, the increase in which is associated with a high risk of breast cancer [60,61,62,63].

Whole grain products, usually abundant in the Mediterranean diet, contain phytic acid, resistant starch, and soluble fiber, which are able to bind and neutralize potentially carcinogenic compounds present in foods [64]. High fiber content of the Mediterranean diet may also exert a protective effect on breast cancer development through a decrease in low-grade chronic inflammation that was previously linked to increased risk [65]. Indeed, following the consumption of dietary fiber, the intestinal microbiota produces short-chain fatty acids (SCFAs), which have the ability to modulate the immune system after being converted into acetyl-CoA, resulting in anti-inflammatory activity, such as the inhibition of the production of inflammatory cytokines, including TNF-α, MCP-1, and IL-6 [66]. The Mediterranean diet is rich in monounsaturated fatty acids (MUFAs) and polyunsaturated fatty acids (PUFAs) which can downregulate the NF-κB protein complex via the peroxisome proliferator-activated receptor (PPAR) and exert anti-inflammatory effects [67]. Omega-3 PUFAs, mainly contained in fish, prevent the conversion of arachidonic acid into proinflammatory eicosanoids such as series 4 leukotrienes and series 2 prostaglandins through substrate competition. They can block excessive inflammatory responses and promote resolution of damage by increasing the clearance of apoptotic cells and debris from inflamed tissues [68]. The results of the present work support the hypothesis that greater adherence to Mediterranean-type dietary pattern may lead to a reduction in breast cancer risk through modulation of the CRP concentration.

Moreover, evidence has accumulated that adhering to a Mediterranean-type diet facilitates weight control [69], in order to combat excess adiposity, which represents a potential risk factor for postmenopausal breast cancer [6].

Strengths of our study include its prospective design, the use of a detailed dietary questionnaire, and the availability of information of several nondietary variables, allowing us to control for their supposed confounding effect in the analysis, and the availability of two plasma samples for a subset of participants, that permitted to evaluate change of inflammatory biomarkers over time. However, some limitations should be taken into account in the interpretation of our results. First, ORDET is an observational study; therefore, it is not possible to find a causal relationship from the results, but only an association that will have to be proven by intervention studies. Second, consumption estimates were based on a single dietary assessment; therefore, we were not able to assess dietary changes, although dietary patterns are more reliable than single foods/nutrients as indicators of long-term usual diet [70]. Moreover, we cannot rule out residual confounding by factors that we were not able to estimate or estimated suboptimally in our questionnaires, such as physical activity and lifetime change in BMI. Another potential limit of our work could be due to the use of the EAT-Lancet index on the participants of the ORDET study: in the 1980s, the concept of environmental sustainability was not taken into consideration as it is today, so the FFQ we used in our study did not include questions relating to this concept. New cohort studies may consider developing a dietary questionnaire tailored to the evaluation of environmental sustainability of the diet. Moreover, collection of both dietary and other lifestyle information at different time points will allow us to evaluate how improving or worsening diet quality over time influence breast cancer risk, ruling out confounding due to other factors and to their change in time. Finally, we recognize that the DASH diet has traditionally been studied for the American population, which has eating habits and dietary assessment instruments that are different from those we used in this work: this may partly explain the lack of association between this diet score and breast cancer risk in our study.

## 5. Conclusions

In conclusion, our results show that adherence to the Mediterranean diet has a protective role against breast cancer development, in particular among postmenopausal women. One possible mechanism explaining its protective role may be through modulation of chronic low-grade inflammation. One of the reasons why the Mediterranean diet, in particular quantified through the IMI, was found to be the dietary model with the greatest protective effect could be partly explained by the fact that this index, built specifically to measure adherence to the Mediterranean Italian diet, may be able to better characterize its healthier aspects compared to the other indices. However, our observational findings should be confirmed by intervention studies. If these results will be confirmed, the adoption of a protective Mediterranean dietary model should be encouraged by means of public health interventions, but also of personalized dietary interventions tailored to individual risk profiles (e.g., postmenopausal women with low-grade inflammation) and preferences in reducing breast cancer incidence and improve prognosis in breast cancer patients. Meanwhile, further studies are warranted to better characterize the effects of a sustainable dietary pattern on breast cancer occurrence.

## Figures and Tables

**Table 1 nutrients-16-01187-t001:** Baseline characteristics of women included in the study by tertile of Italian Mediterranean Index.

Characteristic	Tertile I: Score 0–2 (n = 3526)	Tertile II: Score 3–4 (n = 3690)	Tertile III: Score 5–10 (n = 1850)
Mean ± SD of			
Age, y	48.1 ± 8.3	48.6 ± 8.6	48.3 ± 8.7
Age at menarche, y	12.8 ± 1.6	12.9 ± 1.6	12.9 ± 1.5
Nonalcoholic energy, kcal/d	1621 ± 420	1705 ± 504	1825 ± 506
Body mass index, kg/m^2^	25.3 ± 4.3	25.3 ± 4.4	25.3 ± 4.3
Percentage (n) of			
Menopausal status: Postmenopausal	36.9% (1301)	39.7% (1464)	37. 8% (699)
Perimenopausal	6.0% (212)	5.9% (219)	4.7% (88)
Premenopausal	55.6% (1962)	53.1% (1960)	56.2% (1039)
Not defined	1.5% (51)	1.3% (47)	1.3% (24)
Parity: Nulliparous	11.5% (407)	10.4% (384)	11.9% (220)
1–2 children	66.2% (2334)	65.6% (2419)	65.8% (1217)
>2 children	22.3% (785)	24.0% (887)	22.3% (413)
Age at 1st birth: ≤20 y	3.6% (125)	3.4% (127)	3.5% (65)
>20–≤25 y	37.8% (1332)	37.9% (1398)	35.6% (659)
>25 y	47.1% (1662)	48.3% (1781)	49.0% (906)
n.a. (no children)	11.5% (407)	10.4% (384)	11. 9% (220)
Smoking status: current smokers	20.0% (704	19.2% (707)	20.3% (375)
Ex-smokers	15.0% (531)	14.3% (528)	15.0% (277)
Never smokers	65.0% (2291)	66.5% (2455)	64.7% (1198)
Education (>8 y)	50.5% (1782)	50.0% (1845)	53.6% (992)

**Table 2 nutrients-16-01187-t002:** Hazard ratios (HRs) for developing breast cancer in relation to adherence to the Italian Mediterranean Index, the Greek Mediterranean Index, DASH, and EAT-Lancet score.

	Tertile of Adherence			P-Trend
	I	II	III	
Italian Mediterranean Index				
Score range	0–2	3–4	5–10	
All women				
Cases/Person-years	227/72,975	219/75,218	91/37,122	
HR ^1^	1	0.92 (0.77–1.11125)	0.76 (0.60–0.98)	0.04
HR ^2^	1	0.92 (0.77–1.11)	0.76 (0.60–0.97)	0.04
Postmenopausal women				
Cases/Person-years	85/27,217	100/29,658	29/14,018	
HR ^1^	1	1.07 (0.80–1.43)	0.65 (0.42–0.99)	0.11
HR ^2^	1	1.07 (0.80–1.43)	0.64 (0.42–0.98)	0.11
Premenopausal women				
Cases/Person-years	129/40,377	105/40,268	59/20,923	
HR ^1^	1	0.81 (0.62–1.05)	0.86 (0.63–1.18)	0.22
HR ^2^	1	0.81 (0.62–1.05)	0.87 (0.64–1.19)	0.26
Greek Mediterranean Index				
Score range	0–3	4–5	6–9	
All women				
Cases/Person-years	224/72,677	232/78,331	91/34,306	
HR ^1^	1	1.00 (0.83–1.20)	0.88 (0.69–1.13)	0.37
HR ^2^	1	1.00 (0.83–1.20)	0.88 (0.69–1.13)	0.39
Postmenopausal women				
Cases/Person-years	91/27,976	89/30,042	34/12,876	
HR ^1^	1	0.90 (0.67–1.21)	0.79 (0.53–1.18)	0.24
HR ^2^	1	0.90 (0.67–1.21)	0.78 (0.52–1.16)	0.21
Premenopausal women				
Cases/Person-years	111/39,555	129/42,768	53/19,245	
HR ^1^	1	1.07 (0.83–1.38)	0.96 (0.69–1.34)	0.95
HR ^2^	1	1.07 (0.83–1.39)	0.97 (0.70–1.35)	0.98
DASH diet				
Score range	8–18	19–21	22–32	
All women				
Cases/Person-years	214/73,366	185/56,213	138/55,735	
HR ^1^	1	1.12 (0.92–1.37)	0.84 (0.66–1.04)	0.16
HR ^2^	1	1.13 (0.93–1.38)	0.84 (0.68–1.04)	0.17
Postmenopausal women				
Cases/Person-years	80/27,691	77/21,849	57/21,355	
HR ^1^	1	1.22 (0.89–1.66)	0.91 (0.65–1.29)	0.70
HR ^2^	1	1.22 (0.89–1.68)	0.91 (0.65–1.28)	0.68
Premenopausal women				
Cases/Person-years	121/40,763	100/30,220	72/30,585	
HR ^1^	1	1.11 (0.85–1.45)	0.79 (0.59–1.05)	0.15
HR ^2^	1	1.11 (0.85–1.45)	0.80 (0.59–1.07)	0.18
EAT-Lancet score				
Score range	4–9	10	11–12	
All women				
Cases/Person-years	187/65,858	225/79,870	125/39,586	
HR ^1^	1	0.99 (0.82–1.21)	1.10 (0.87–1.38)	0.48
HR ^2^	1	1.00 (0.82–1.21)	1.10 (0.88–1.39)	0.44
Postmenopausal women				
Cases/Person-years	76/25,448	97/30,357	41/15,090	
HR ^1^	1	1.07 (0.79–1.44)	0.90 (0.61–1.31)	0.69
HR ^2^	1	1.06 (0.78–1.43)	0.89 (0.61–1.30)	0.63
Premenopausal women				
Cases/Person-years	99/36,032	117/44,060	77/21,476	
HR ^1^	1	0.97 (0.74–1.27)	1.30 (0.96–1.75)	0.13
HR ^2^	1	0.98 (0.75–1.28)	1.29 (0.96–1.75)	0.13

^1^ Adjusted for age and nonalcoholic energy intake. ^2^ Further adjusted for age at menarche (continuous), menopausal status (premenopausal/perimenopausal/postmenopausal/not classified), parity (nulliparous/1–2 sons/>2 sons), age at first birth (≤20 years/>20–≤25 years/>25 years), smoking status (never/former /current), education (≤8 years, >8 years), and BMI (continuous).

**Table 3 nutrients-16-01187-t003:** Coefficient (β) for the inflammatory markers C-reactive protein (CRP), TNF-α, IL-6, leptin, and adiponectin in relation to adherence to the Italian Mediterranean Index, Greek Mediterranean Index, DASH, and EAT-Lancet score.

	C-Reactive Protein [CRP]	TNF-α	IL-6	Leptin	Adiponectin
β ^1^ (Confidence Interval 95%)					
Italian Mediterranean Index	−0.10 (−0.18, −0.02)	0.05 (−0.08, 0.19)	0.02 (−0.11, 0.15)	0.01 (−0.07, 0.09)	0.44 (−0.02, 0.11)
Greek Mediterranean Index	−0.13 (−0.21, −0.04)	0.04 (−0.09, 0.18)	0.01 (−0.12, 0.14)	0.04 (−0.03, 0.12)	0.03 (−0.03, 0.1)
DASH diet	0.01 (−0.08, 0.09)	−0.01 (−0.14, 0.13)	−0.03 (−0.16, 0.10)	0.06 (−0.02, 0.14)	−0.03 (−0.09, 0.04)
EAT-Lancet score	0.10 (0.02, 0.18)	−0.03 (−0.17, 0.11)	−0.06 (−0.20, 0.07)	0.00 (−0.07, 0.08)	−0.04 (−0.11, 0.02)

^1^ Adjusted for age, menopausal status, and the time gap between recruitment and conduct of the nested case–control study.

## Data Availability

The data presented in this study are available on request from the principal investigator due to ethical reasons: the Ethical Committee does not allow open/public sharing of data pertaining to individuals.

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
