# Peer review of "Adherence to Diet Quality Indices and Breast Cancer Risk in the Italian ORDET Cohort"

_nutrients, 2024, doi:10.3390/nu16081187_

Round 1
Reviewer 1 Report
Comments and Suggestions for Authors
This manuscript compelling exploration of the relationship between dietary patterns and breast cancer (BC) risk, with a particular focus on chronic low-grade inflammation as a potential mediating mechanism. The use of the ORDET cohort and the detailed analysis of dietary adherence provide valuable insights into this critical area of research. Below are some comments further enhancing the clarity, robustness, and impact of your work:
The manuscript is well-written and logically structured. However, it would benefit from a more detailed explanation of the selection criteria for the dietary patterns examined. Additionally, clarifying the rationale behind focusing on these specific diets and their relevance to the study population could enhance the reader's understanding of your research choices.
The inverse association between the Italian Mediterranean Index (IMI) and BC risk, particularly among postmenopausal women, is intriguing. Expanding on the biological plausibility of how adherence to the IMI might modulate chronic inflammation and subsequently reduce BC risk could make this section more compelling. Incorporating recent literature that supports or contradicts these findings would also provide a more comprehensive context for your results.
While you have briefly touched upon the limitations of your study, a more detailed discussion would enhance the manuscript. For instance, considering the observational nature of the study, discussing the potential for residual confounding and how it might affect the interpretation of your results is crucial. Additionally, reflecting on the generalizability of your findings to populations outside the ORDET cohort would be valuable.
Provide a more detailed discussion on the distinct components of the Italian Mediterranean diet (IMI), the Greek Mediterranean Index, the DASH diet, and the EAT-Lancet diet. Explaining how these dietary patterns differ in terms of food composition, nutrient profiles, and potential health effects would enrich the readers' understanding and highlight the relevance of your study in the context of current dietary recommendations.
While you have proposed chronic low-grade inflammation as a potential mechanism linking dietary patterns to breast cancer risk, consider discussing other plausible mechanisms supported by existing literature. For example, the role of hormonal regulation, oxidative stress, and metabolic pathways in mediating the relationship between diet and cancer risk could be explored further. Integrating insights from mechanistic studies could strengthen the biological plausibility of your findings.
Discuss the clinical implications of your findings in the context of breast cancer prevention and management. How can healthcare professionals use this information to counsel patients on dietary choices? Consider addressing the potential role of personalized dietary interventions tailored to individual risk profiles and preferences in reducing breast cancer incidence and improving outcomes.
By addressing these additional comments, the manuscript can become even more comprehensive, impactful, and accessible to readers in the field of breast cancer research and nutritional epidemiology.
Reviewer 2 Report
Comments and Suggestions for Authors
The paper provides valuable information about the association of dietary patterns and breast cancer risk studying the ORDET cohort.
The introduction presents the significance of breast cancer as a global health concern and the importance of diet in cancer prevention. It also provides an overview of previous research on dietary patterns and breast cancer risk.
I have the following comments for the authors:
-Studies like this provide valuable data, however they cannot establish causation, as other factors like genetics, environmental exposure, etc, also play significant roles in cancer development.
-It is important to explain the rationale for the exclusion criteria. For example, patients with autoimmune diseases were not excluded, but these patients usually have elevated serum inflammation markers.
- Moreover, it would be beneficial to discuss the rationale for selecting these specific biomarkers and their relevance to chronic low-grade inflammation and breast cancer risk.
- Consider providing a clearer and more analytical report of the results by dividing into subsections corresponding to each analysis (associations between diet scores and breast cancer risk, associations with inflammatory markers etc). The main finding is the association between adherence to the Italian Mediterranean Index (IMI) and reduced breast cancer risk, especially among postmenopausal women. However, the authors should also interpret the findings regarding the Greek Mediterranean Index, DASH diet, and EAT-Lancet score by analyzing possible reasons for the lack of association. It's quite impressive that association was found primarily for Italian Mediterranean Index (IMI) in postmenopausal women. It might be worth discussing potential reasons for this specificity and exploring whether there are biological or lifestyle factors influencing this outcome. Dietary patterns have complex relationships with health outcomes, and different dietary indices may capture varying aspects of diet quality or composition. Please also evaluate factors like estrogen metabolism, anti-inflammatory properties, antioxidant effects etc.
- Discuss any discrepancies between the current study and published literature and provide explanations based on methodological differences, population characteristics etc
- The discussion analyzes potential biological mechanisms underlying the protective effects of the Mediterranean diet against breast cancer, particularly through modulation of chronic low-grade inflammation. To strengthen this aspect,mention more details about pathways involved, supported by relevant literature. In addition, the study emphasizes the promotion of the Mediterranean diet as a preventive measure against breast cancer. Consider expanding public health interventions and policy initiatives for healthy dietary patterns. The discussion also acknowledges the limitations of the study; consider discussing potential avenues to address these limitations.
Comments on the Quality of English Language
minor editing
Round 2
Reviewer 2 Report
Comments and Suggestions for AuthorsΤhe authors addressed the issues I raised and improved the text, particularly in the discussion section with the additions they made.
I noticed is that there is some error in the numbering of the references, specifically with 1 and 2 in the Introduction.
Comments on the Quality of English Languageminor editing